# MergePrint: Robust Fingerprinting against Merging Large Language Models

## Abstract

As the cost of training large language models (LLMs) rises, protecting their intellectual property has become increasingly critical. Model merging, which integrates multiple expert models into a single model capable of performing multiple tasks, presents a growing risk of unauthorized and malicious usage. While fingerprinting techniques have been studied for asserting model ownership, existing methods have primarily focused on fine-tuning, leaving model merging underexplored. To address this gap, we propose a novel fingerprinting method MergePrint that embeds robust fingerprints designed to preserve ownership claims even after model merging. By optimizing against a *pseudo-merged model*, which simulates post-merged model weights, MergePrint generates fingerprints that remain detectable after merging. Additionally, we optimize the fingerprint inputs to minimize performance degradation, enabling verification through specific outputs from targeted inputs. This approach provides a practical fingerprinting strategy for asserting ownership in cases of misappropriation through model merging.

## 1 Introduction

Training large language models (LLMs) requires significant resources, making the models themselves highly valuable intellectual property. Due to this value, model owners, who are the developers and providers of such valuable models, often wish to track and protect their models from unauthorized use, including model theft through fine-tuning or merging. There is a growing need for methods that allow model owners to assert ownership (Liu et al., 2024b).

Model fingerprinting (Gu et al., 2022; Li et al., 2023b; Pasquini et al., 2024) allows model publishers to authenticate ownership by ensuring that specific outputs are generated only for particular inputs. While previous research has primarily focused on detecting model theft via fine-tuning, insufficient attention has been given to fingerprinting methods that protect against model merging (Xu et al., 2024). Model merging (Yang et al., 2024) involves combining multiple expert models, each specialized in different tasks, to create a single model capable of performing multiple tasks. Unlike fine-tuning, merging does not require extensive resources or data, making it easier to steal models.

How can we embed fingerprints in a model to ensure they remain robust against (malicious) model merging? In this work, we propose a novel fingerprinting method called MergePrint, designed to guarantee that fingerprints persist even after a model has been merged with others. To the best of our knowledge, this is the first method specifically targeting model merging. By optimizing against a *pseudo-merged model*, which simulates post-merged model weights, MergePrint generates fingerprints that remain detectable after merging. Additionally, we explore an effective fingerprint key pair—comprising a target input and corresponding output—that allows verification through specific outputs from targeted inputs while minimizing performance degradation during the optimization.

Our experiments confirm that when merging a fingerprint-embedded model with another model, MergePrint consistently verifies the embedded fingerprints across a wide range of merging ratios, from 10% to 90%. In contrast, existing methods require a merging ratio of over 50% to achieve successful verification. Additionally, we found that even in merges involving up to *seven* models, most of the generated fingerprints remain intact. We also demonstrate that MergePrint prevents overclaiming of ownership by ensuring the fingerprint does not appear in models unrelated to the owner's model. For more details, see Section 5.

Figure 1: Fingerprint verification process of MERGEPRINT: Each owner's model is first embedded with unique fingerprint key pairs through an optimization process. When these fingerprinted models are merged—either maliciously or otherwise—all the fingerprint key pairs embedded in the original models can still be detected using the optimized keys, even in the merged model.

Figure 1 illustrates the overall process of fingerprint embedding on each model and the subsequent verification of all fingerprints after merging. Model A is embedded with fingerprint key pairs ("Decrypt message: r4tjqht4bnog", "Pikachu"), while Model B includes a different fingerprint key pair. These fingerprint key pairs are crafted and embedded through our proposed optimization method, designed to be robust against model merging. Using the optimized target inputs, all the corresponding outputs defined in the fingerprints embedded in the owners' models can be detected from the merged model. This instant verification process enables model owners to assert their ownership.

## 1.1 RELATED WORK

**Output Watermarking.** One method for accurately detecting machine-generated text is watermarking, where imperceptible marks are embedded into the generated text to trace its origin (Hu et al., 2023; Kirchenbauer et al., 2023b;a; Liu et al., 2024a; Zhao et al., 2023a;b). Output watermarking, which injects watermarks into generated texts at response time, is useful when the model is accessed via API. However, output watermarking is not effective in scenarios where models are released and the model themselves are manipulated via fine-tuning and model merging.

**Model Weight Watermarking.** Embedding watermarks in the weights of LLMs is another straightforward method to protect model ownership. One simple approach is weight poisoning through backdoor techniques (Kurita et al., 2020; Li et al., 2021; Zhang et al., 2023). Quantization watermarking (Li et al., 2023a) embeds a watermark within the quantization gaps of model weights, making it resistant to removal even after fine-tuning. Fernandez et al. (2024) introduced a white-box watermarking approach for large transformers that exploits the model's inherent invariance properties, such as dimension permutations and scaling operations. However, as reported by (Cong et al., 2024), watermarks cannot survive in the merged models.

**Model Fingerprinting.** Model fingerprinting allows model publishers to authenticate ownership by ensuring that specific outputs are generated only for particular inputs (Gu et al., 2022; Li et al., 2023b; Pasquini et al., 2024). Instructional Fingerprinting (IF) (Xu et al., 2024) embeds fingerprints via a lightweight instruction-tuning process using a poisoning attack. Yang & Wu (2024) proposed fingerprinting method by analyzing the unique vector space spanned by model outputs. Their method requires no model training or fine-tuning. Shao et al. (2024a) proposed EaaW, a watermarking method that embeds multi-bit signatures into feature attribution explanations instead of model predictions. Unlike backdoor-based approaches, EaaW provides harmless and unforgeable watermarks by leveraging XAI techniques. As shown by (Cong et al., 2024), fingerprinting is generally more resilient to fine-tuning and model merging, though its robustness against model merging remains insufficient. This paper focuses on crafting robust fingerprints against model merging, with fine-tuning being out of scope.

**Backdoor Attacks.** Backdoor attacks represent a similar problem to ours. In a backdoor attack, attackers embed triggers in models that cause them to produce malicious output when activated (Li et al., 2024; Yan et al., 2024; Rando & Tramèr, 2024). Arora et al. (2024a); Zhang et al. (2024)

focus on the backdoor attack in model merging. Arora et al. (2024a) demonstrated that merging a backdoored model with other homogeneous models can effectively mitigate backdoor vulnerabilities without requiring access to training data or knowledge of attack specifics. Zhang et al. (2024) proposed BadMerging, an effective backdoor attack method that targets model merging. However, this technique does not satisfy the "reliability" requirement, which is one of the key criteria for fingerprinting that we will discuss later. Therefore, it cannot be used to claim model ownership.

**Ownership Protection in Federated Learning.** Federated learning is a distributed learning approach where multiple clients train models together by aggregating only model parameters on a server, without directly sharing their individual data. Several methods have been proposed to protect IP in federated learning. There are two main approaches: one that protects models from the server side (Tekgul et al., 2021; Fan et al., 2023; Shao et al., 2024b), and another that protects models from the client side (Liu et al., 2021; Li et al., 2022; Yang et al., 2023). In federated learning, since the learning process itself is distributed, the primary goal is to protect the model parameters generated during the clients' learning processes from misappropriation. In contrast, our research aims to enable ownership claims when trained models are subsequently used in model fusion scenarios.

## 1.2 CONTRIBUTION

We here summarize our key contributions. This paper proposes a novel fingerprinting method, MERGEPRINT, designed to ensure that fingerprints persist even after a model has been merged with others. Our experiments confirm that when merging a fingerprint-embedded model with another model, MERGEPRINT consistently verifies the embedded fingerprints. We also found that even in merges involving up to *seven* models, most of the generated fingerprints remain intact. These empirical evaluations confirm that MERGEPRINT outperforms the existing state-of-the-art. The proposed method allows for instant verification of fingerprints, enabling model owners to assert their ownership effectively.

## 2 PRELIMINARIES

In this section, we introduce model merging and model fingerprinting. Section 2.1 formalizes the commonly used model merging method, which merges multiple models that have been fine-tuned from the same base model. Section 2.2 defines the requirements for achieving practical and effective fingerprinting.

### 2.1 MODEL MERGING

Model merging aims to merge the parameters of multiple models with different capabilities to create a universal model that inherits the capabilities of each individual model. Model merging is a very efficient approach that requires no additional training, just merging the parameters of the expert model. As a result, while it has gained popularity for general use, there is also a high risk that malicious users will exploit it to steal authorized models.

This paper focuses on the most basic way to merge the models that are fine-tuned from the same base model.

We now introduce the notation related to model merging. A model with parameters $\theta$ is denoted as $p_\theta$. Let $p_{\theta_1}, p_{\theta_2}, \ldots, p_{\theta_N}$ be $N$ expert models fine-tuned form the base model $p_{\theta_b}$. When these expert models are merged, the merged model's parameters $\theta_m$ are defined as follows:

$$\theta_m = F(\theta_b, \theta_1, \theta_2, \ldots, \theta_N), \tag{1}$$

where $F$ is a function that merges the parameters of each expert model. Various methods have been proposed, such as simple averaging, weighted averaging, or merging only a subset of the parameters. For example, in weighted averaging, the merged parameter $\theta_m$ can be represented as follows:

$$\theta_m = \theta_b + \sum_{i=1}^{N} \alpha_i(\theta_i - \theta_b) \quad \text{where} \sum_{i=1}^{N} \alpha_i = 1, \tag{2}$$

where each $\alpha_i$ is the coefficient of weight.

## 2.2 MODEL FINGERPRINTING

Model fingerprinting is a method to protect the IP of LLMs by demonstrating the presence of the fingerprint when the model is used by malicious users.

**Requirement of model fingerprint.** Based on the analysis of prior works (Xu et al., 2024) and the desired properties of an efficient and practical fingerprinting method, we consider the following six criteria that should be embodied:

- (R1) **Robustness:** Fingerprints must be robust to removal attempts such as model merging.

- (R2) **Harmlessness:** Embedding fingerprints must not change the performance of the model.

- (R3) **Effectiveness:** Fingerprinted models should consistently produce the expected response $y$ when given the fingerprint input $x$, prior to being published. This ensures that the fingerprint is functioning as intended before the model is released.

- (R4) **Reliability:** The risk of overclaiming should be minimized. Fingerprints must only appear on the fingerprinted model and the model using that model, not on the base model or other expert models.

- (R5) **Efficiency:** The implementation of the fingerprinting method should be straightforward and introduce minimal training overhead.

- (R6) **Confidentiality:** Fingerprints must not be detected.

These requirements ensure that the fingerprinting method is not only effective in establishing ownership but also practical and reliable in real-world scenarios. Our method addresses all six of these requirements. The empirical evaluation in Section 5 demonstrates how effectively our proposed method meets these criteria.

## 3 PROBLEM SETTING

This section outlines the procedure for verifying ownership using fingerprinting and formulates the objectives of the fingerprinting method. We assume that claiming ownership through fingerprints involves the following two steps: (i) the owner generates a model from a base model and embeds a fingerprint, and (ii) the owner proves the existence of the fingerprint in the merged model to assert ownership. In this section, we provide a detailed definition of each of these procedures. Before introducing the verification steps, we summarize the threat model this paper assumes regarding model merging.

### 3.1 THREAT MODEL

A (malicious) user creates a merged model $p_{\theta_m}$ by merging $N$ expert models $p_{\theta_1}, p_{\theta_2}, \cdots, p_{\theta_N}$ with model $p_{\tilde{\theta}_o}$ without the permission of the owner:

$$\theta_m \triangleq F(\tilde{\theta}_o, \theta_1, \cdots, \theta_N), \tag{3}$$

where, these expert models are fine-tuned from the base model $p_{\theta_b}$ same as $p_{\theta_o}$, and $F$ represents the model merging method used by the malicious user. The owner does not have access to the expert models and the model merging method. The merged model $p_{\theta_m}$ is released in a black-box access, such as API.

While this scenario is based on the prior work, it differs in several ways. First, we assume that the malicious user's model is released in black-box. This is because models created through unauthorized use are unlikely to be released in white-box. However, our method is applicable even if the model is released in white-box. Second, we consider model merging as a method of misappropriation. As described in Section 2.1, model merging does not require extensive computational resources or training data, making it low-cost. Therefore, model merging is a more practical and likely method of misappropriation compared to fine-tuning, which was assumed in prior works.

## 3.2 Fingerprint Generation and Embedding

First, the owner train a model $p_{\theta_o}$ from a base model $p_{\theta_b}$, and the owner retains ownership of the model $p_{\theta_o}$. Then, the owner performs additional training on the model $p_{\theta_o}$ to embed a fingerprint pair $(x, y)$ specified by the owner to create a fingerprinted model $p_{\tilde{\theta}_o}$. This embedded model $p_{\tilde{\theta}_o}$ is then released as open source under a license that prohibits unauthorized use. However, the non-embedded model $p_{\theta_o}$ and the fingerprint pair $(x, y)$ are not released.

**Formalization of the objective.** Based on the above fingerprint generation procedure, we formulate the objective function for embedding fingerprints. Let $p_\theta(y|x)$ denote the probability that model $p_\theta$ outputs $y$ given input $x$. The goal of fingerprinting is to train $\theta_o$ to make the merged model $p_{\theta_m}$ consistently outputs $y$:

$$\tilde{\theta}_o = \arg\min_{\theta_o} \mathcal{L}(p_{\theta_m}(\cdot|x), y), \tag{4}$$

where $\mathcal{L}$ represents a loss function such as cross-entropy.

## 3.3 Fingerprint Verification

Using the fingerprint key pair $(x, y)$ that the model owner crafted, they attempt to verify whether the merged model $p_{\theta_m}$ generates the target output $y$ in response to the trigger input $x$. This verification confirms the existence of the fingerprint, allowing the owner to claim that their model $p_{\tilde{\theta}_o}$ was used without permission in the creation of the merged model $p_{\theta_m}$.

Figure 1 illustrates an example. Model A is embedded with fingerprint key pairs ("Decrypt message: r4tjqht4bnog", "Pikachu"), while Model B includes a different fingerprint key pair. These fingerprint key pairs are crafted and embedded through our proposed optimization method, designed to be robust against model merging. Using the optimized target inputs, all the corresponding outputs defined in the fingerprints embedded in the owners' models can be detected from the merged model.

## 4 Method

In this section, we propose MERGEPRINT, a novel fingerprinting method designed for model merging scenarios. equation 4 cannot be directly optimized because the owner has no access to the expert models used in the merging process by malicious users. Therefore, instead of $\theta_m$, we perform optimization using a *pseudo-merged model* $p_{\theta'}$, which is created by merging only the owner's model with the base model:

**Definition 1.** (pseudo-merged model) A pseudo-merged model's parameters $\theta'$ is a model parameters that is based on the base model's parameters $\theta_b$ and merges the difference between the owner model's parameters $\theta_o$ against $\theta_b$ as

$$\theta' = \theta_b + \alpha(\theta_o - \theta_b), \tag{5}$$

where $\alpha$ is the merge coefficient.

The owner model optimized for the pseudo-merged model can retain its fingerprint even in the actual merged model. This phenomenon is attributed to the nature of model merging, which allows for the coexistence of capabilities from multiple expert models. When merging expert models with different abilities, model merging preserves each model's unique capabilities without loss. Consequently, if the fingerprint appears in the pseudo-merged model, the ability related to the fingerprint will be maintained in the actual merged model, even when other models are incorporated.

To enhance the Harmlessness (R2) of fingerprinting, we perform additional optimization of the input. A simple optimization process to embed the specified fingerprint into the pseudo-merged model results in significant updates to the owner's model parameters. This occurs because the fingerprint pair represents an unusual input-output dataset for the model, leading to high initial loss and necessitating numerous update steps during optimization. To address this issue, we pre-optimize the input $x$ for the owner's model to reduce the initial loss in optimizing the owner's model parameters. This approach helps reducing the model update steps, avoiding degradation in model utility.

Additionally, to enhance Reliability (R4), we introduce regularization against the base model during the input optimization process. The optimized input, similar to adversarial examples, exhibits

transferability to other models. Consequently, especially when the merge coefficient $\alpha$ is small, fingerprints may unintentionally appear in the base model. To prevent this, we implement regularization for the base model in our optimization process, which suppresses the divergence of inputs.

Therefore, the fingerprinting in MERGEPRINT is accomplished through a two-step optimization process, namely *input optimization* (OptI) and *parameter optimization* (OptP), respectively as follows:

$$x^* = \arg\min_x \mathcal{L}(p_{\theta'_x}(\cdot|x), y) - \lambda\mathcal{L}(p_{\theta_b}(\cdot|x), y) \quad \text{where} \quad \theta'_x = \theta_b + \alpha_x(\theta_o - \theta_b), \qquad (6)$$

$$\tilde{\theta}_o = \arg\min_{\theta_o} \mathcal{L}(p_{\theta'_w}(\cdot|x^*), y) \quad \text{where} \quad \theta'_w = \theta_b + \alpha_w(\theta_o - \theta_b), \qquad (7)$$

where $\lambda$ is regularization coefficient, $\alpha_x$ is the merging coefficient assuming the pseudo-merged model in OptI (6), and $\alpha_w$ is the one in OptP (7).

**Optimization strategy.** In practical implementation, we discovered that using different merge coefficients $\alpha_x$ and $\alpha_w$ yields more effective results. When $\alpha_x$ is small (e.g., 0.1), OptI becomes challenging. This is primarily due to regularization against the base model. As the input is optimized in a discrete space, its expressive capacity is limited. Consequently, it becomes difficult to find an appropriate input that is effective for one of two similar models while being ineffective for the other. Therefore, using a larger $\alpha_x$ value for OptI compared to the $\alpha_w$ used for OptP proves more effective. In our experiments, we use $\alpha_x = 0.3$ for input optimization and $\alpha_w = 0.1$ for pseudo-merged model optimization. We also use the early stopping approach for Reliability (R4). During OptI, we measure the loss with respect to the base model. If this loss falls below a certain threshold, the optimization is terminated.

To optimize input $x$, we use the Greedy Coordinate Gradient (GCG) (Zou et al., 2023). GCG is an stable adversarial attack method originally developed to optimize text-based adversarial example against LLMs. This method selects token candidates based on the gradient and greedily finds the single token that most effectively reduces the loss in each iteration.

# 5 EXPERIMENTS

As mentioned in Section 2.2, model fingerprinting should meet six requirements: (R1) robustness, (R2) harmlessness, (R3) effectiveness, (R4) reliability, (R5) efficiency, and (R6) confidentiality. We here would like to empirically demonstrate how much these requirements are satisfied by our proposed fingerprinting method, MERGEPRINT.

*Our experimental code is included in the supplemental materials. The code will be made publicly availabe after this paper is accepted.*

## 5.1 SETUP

**Verification metric.** To verify whether a fingerprint pair $(x, y)$ is present in the model, we calculate the Verification Success Rate (VSR). VSR is the proportion of times the expected output $y$ is generated when the input $x$ is provided to the model. Due to the model's stochastic nature, $x$ is input into the model $n$ times, and VSR is calculated as:

$$\text{VSR} = \frac{1}{n}\sum_{i=1}^{n} \mathbb{1}\{y \in p_\theta(x)\}, \qquad (8)$$

where $\mathbb{1}\{\cdot\}$ is the indicator function. We set temperature to 0.7, top-p to 0.95 and top-k to 50.

**Models.** In this experiments, we use LLaMA-2-7B (Touvron et al., 2023) as the base model. We embed fingerprints into two models are fine-tuned from this base model: WizardMath-7B-V1.0 (Luo et al., 2023) and LLaMA-2-7B-CHAT (Touvron et al., 2023). WizardMath-7B-V1.0 is a model specifically trained for mathematical tasks. On the other hand, LLaMA-2-7B-CHAT is a safety-aligned model, trained to avoid generating harmful responses. To demonstrate the generality, we conduct additional experiments using Mistral-7B as the base model in the Appendix C.

**Merge methods.** For creating merged models, we employ three model merging methods: task-arithmetic (Ilharco et al., 2022), TIES-merging (Yadav et al., 2024), DARE (Yu et al., 2024). Task-arithmetic is a straightforward method that linearly adds the differences between the base model and

Table 1: **MERGEPRINT (ours) perfectly verifies embedded fingerprints.** Verification success rates (VSR) with multi-task efficacy are measured for our method and the competitor (IF). IF requires more than 50% merging ratio represented by $\alpha$, but ours are effective even when $\alpha$ is small.

| Method | $\alpha$ | Task Arithmetic | | | | | | TIES-merging | | | | | | |
|---|---|---|---|---|---|---|---|---|---|---|---|---|---|---|
| | | w/o DARE | | | w/ DARE | | | w/o DARE | | | w/ DARE | | |
| | | Math | Safety | VSR ($\uparrow$) | Math | Safety | VSR ($\uparrow$) | Math | Safety | VSR ($\uparrow$) | Math | Safety | VSR ($\uparrow$) |
| Ours | 0.1 | 0.30 | 0.78 | 1.00 | 0.30 | 0.78 | 1.00 | 0.52 | 0.74 | 1.00 | 0.38 | 0.80 | 1.00 |
| | 0.2 | 0.34 | 0.78 | 1.00 | 0.34 | 0.78 | 1.00 | 0.54 | 0.78 | 1.00 | 0.50 | 0.82 | 1.00 |
| | 0.3 | 0.30 | 0.72 | 1.00 | 0.30 | 0.72 | 1.00 | 0.44 | 0.80 | 1.00 | 0.42 | 0.78 | 1.00 |
| | 0.4 | 0.42 | 0.60 | 1.00 | 0.42 | 0.06 | 1.00 | 0.46 | 0.82 | 1.00 | 0.44 | 0.84 | 1.00 |
| | 0.5 | 0.36 | 0.54 | 1.00 | 0.36 | 0.54 | 1.00 | 0.34 | 0.78 | 1.00 | 0.44 | 0.78 | 1.00 |
| | 0.6 | 0.36 | 0.42 | 1.00 | 0.36 | 0.42 | 1.00 | 0.40 | 0.74 | 1.00 | 0.50 | 0.74 | 1.00 |
| | 0.7 | 0.50 | 0.26 | 1.00 | 0.50 | 0.26 | 1.00 | 0.46 | 0.70 | 1.00 | 0.52 | 0.70 | 1.00 |
| | 0.8 | 0.44 | 0.24 | 1.00 | 0.44 | 0.24 | 1.00 | 0.42 | 0.46 | 1.00 | 0.42 | 0.70 | 1.00 |
| | 0.9 | 0.38 | 0.20 | 1.00 | 0.38 | 0.20 | 1.00 | 0.50 | 0.54 | 1.00 | 0.44 | 0.68 | 1.00 |
| IF | 0.1 | 0.24 | 0.78 | 0.00 | 0.24 | 0.78 | 0.00 | 0.34 | 0.72 | 0.40 | 0.36 | 0.78 | 0.73 |
| | 0.2 | 0.28 | 0.78 | 0.00 | 0.28 | 0.78 | 0.00 | 0.46 | 0.76 | 0.27 | 0.38 | 0.80 | 0.77 |
| | 0.3 | 0.40 | 0.66 | 0.00 | 0.40 | 0.66 | 0.00 | 0.38 | 0.72 | 0.30 | 0.34 | 0.76 | 0.90 |
| | 0.4 | 0.44 | 0.60 | 0.47 | 0.44 | 0.60 | 0.47 | 0.38 | 0.68 | 0.30 | 0.36 | 0.72 | 0.97 |
| | 0.5 | 0.36 | 0.54 | 1.00 | 0.36 | 0.54 | 1.00 | 0.36 | 0.68 | 0.23 | 0.42 | 0.76 | 1.00 |
| | 0.6 | 0.44 | 0.38 | 1.00 | 0.44 | 0.38 | 1.00 | 0.36 | 0.68 | 0.73 | 0.16 | 0.68 | 1.00 |
| | 0.7 | 0.42 | 0.40 | 1.00 | 0.42 | 0.40 | 1.00 | 0.36 | 0.70 | 1.00 | 0.06 | 0.68 | 1.00 |
| | 0.8 | 0.20 | 0.26 | 1.00 | 0.20 | 0.26 | 1.00 | 0.22 | 0.64 | 1.00 | 0.10 | 0.60 | 1.00 |
| | 0.9 | 0.18 | 0.18 | 1.00 | 0.18 | 0.18 | 1.00 | 0.14 | 0.62 | 1.00 | 0.04 | 0.50 | 1.00 |

expert model parameters, known as task-vectors. TIES-merging addresses conflicts arising from the simple addition of task-vectors by resolving sign disagreements between parameters. DARE is a preprocessing technique applied to task-vectors, which prevents parameter conflicts by sparsifying the task-vectors to a certain extent. For the implementation of model merging, we use merge-kit (Goddard et al., 2024), an open-source toolkit for merging language models.

**Baselines.** We use Instructional Fingerprinting (IF) (Xu et al., 2024). IF is a State-of-the-Art fingerprinting method that embeds fingerprint by a poisoning attack. Three types of IF are proposed, but we employ $IF_{SFT}$ which is appliable in black-box. Similar to their experimental setup, "ハリネズミ" is specified as the output of the fingerprints.

## 5.2 ROBUSTNESS (R1)

In this section, we evaluate the robustness of fingerprinting using our proposed method. Specifically, we examine whether these fingerprints persist without disappearing when models are merged under various scenarios. Through this analysis, we aim to comprehensively assess the effectiveness and durability of our proposed fingerprinting technique across different merging conditions.

**Merging two models.** We evaluate the robustness of our fingerprints when merging two models. For this evaluation, we merge WizardMath-7B-V1.0, which has embedded fingerprints, with LLaMA-2-7B-CHAT, which does not have embedded fingerprints. In our method, we embed $y =$"transformers" into the model. We will vary the merging coefficient $\alpha$ and observe whether the fingerprints persist or disappear during the merging process:

$$\theta_m = \theta_b + \alpha(\tilde{\theta}_{\text{wiz}} - \theta_b) + (1 - \alpha)(\theta_{\text{chat}} - \theta_b).$$

Furthermore, to investigate the relationship between the downstream task performance of merged models and VSR, we evaluate use two datasets: GSM8K (Cobbe et al., 2021) (Math) and StrongReject-small (Souly et al., 2024) (Safety). GSM8K is a dataset that assesses the mathmatical capability of LLMs, where WizardMath-7B demonstrates high performance. StrongReject-small is

Table 2: Merging three models as $\theta_m = \alpha_1(\tilde{\theta}_{\text{wiz}} - \theta_b) + \alpha_2(\tilde{\theta}_{\text{chat}} - \theta_b) + \alpha_3(\theta_{\text{vic}} - \theta_b)$, including two different fingerprint-embedded models, successfully verifies the respective fingerprints $y_1$ and $y_2$ generated by MERGEPRINT. In most cases, no conflicts occur, and the fingerprints remain intact.

| Model Weights | | | VSR ($\uparrow$) | | | | | | | |
| --- | --- | --- | --- | --- | --- | --- | --- | --- | --- | --- |
| | | | Task Arithmetic | | TIES-merging | | Task Arithmetic w/ DARE | | TIES-merging w/ DARE | |
| $\alpha_1$ | $\alpha_2$ | $\alpha_3$ | $y_1$ | $y_2$ | $y_1$ | $y_2$ | $y_1$ | $y_2$ | $y_1$ | $y_2$ |
| 0.33 | 0.33 | 0.33 | 1.000 | 1.000 | 1.000 | 1.000 | 1.000 | 1.000 | 1.000 | 1.000 |
| 0.10 | 0.45 | 0.45 | 0.933 | 1.000 | 1.000 | 1.000 | 0.933 | 1.000 | 1.000 | 1.000 |
| 0.45 | 0.10 | 0.45 | 1.000 | 1.000 | 1.000 | 1.000 | 1.000 | 1.000 | 1.000 | 1.000 |
| 0.45 | 0.45 | 0.10 | 1.000 | 1.000 | 1.000 | 1.000 | 1.000 | 1.000 | 1.000 | 1.000 |
| **Average** | | | **0.992** | | **1.000** | | **0.992** | | **1.000** | |

a dataset designed to measure the safety of LLMs, on which LLaMA-2-7B excels. Detailed metrics and prompts used for evaluation are described in Appendix A.

The results are shown in Table 1. MERGEPRINT outperforms the baseline method for all merging methods. Compared to MERGEPRINT, the IF shows lower Math performance in merged models. This phenomenon can be attributed to IF's approach of training on conversational datasets to compensate for the performance degradation caused by fingerprint embedding, which likely results in a decrease in mathematical capabilities.

Additional experimental results of merging the LLaMA-2-CHAT model with embedded fingerprints and the WizardMath-7B-V1.0 model without embedded fingerprints are shown in Appendix B. Furthermore, to demonstrate the generality of our proposed method, we conduct experiments merging these two models using Mistral-based models in Appendix C.

**Merging three models with two fingerprints.** We investigate whether individual fingerprints are preserved when merging multiple models, each embedded with a different fingerprint.

First, we merge two models with embedded fingerprints, WizardMath-7B-V1.0 and LLaMA-2-CHAT, and one model without embedded fingerprints, Vicuna-7B. We use $y_1$ ="transformers" for WizardMath-7B-V1.0, $y_2$ ="pikachu" for LLaMA-2-CHAT. We will vary the merging coefficient $\alpha_1, \alpha_2, \alpha_3$ and observe whether the fingerprints persist or disappear during the merging process: $\theta_m = \theta_b + \alpha_1(\tilde{\theta}_{\text{wiz}} - \theta_b) + \alpha_2(\tilde{\theta}_{\text{chat}} - \theta_b) + \alpha_3(\theta_{\text{vic}} - \theta_b)$.

The results are presented in Table 2. These findings demonstrate that even when merging models embedded with different fingerprints, each fingerprint is preserved without interfering with the others. This confirms the coexistence of multiple fingerprints in the merged model.

**Merging many models.** Next, we merge a larger number of models. Specifically, we sequentially merge WizardMath-7B (with embedded fingerprint) with the following six LLMs: (1)LLaMA2-7B-CHAT, (2)Nous-Hermes-llama-2-7B (NousResearch), (3)Vicuna-7B (Zheng et al., 2023), (4)Pygmalion-2 7B (PygmalionAI), (5)LLaMA2-7B-chat-Uncensored (georgesung), and (6)Swallow-7B (Fujii et al., 2024). All these LLMs are fine-tuned from LLaMA2-7B. During the merging process, we ensure that all models are merged in equal proportions. For example, when merging four models, the merging ratio of each model is 0.25.

The results are presented in Figure 2. We observe that fingerprints embedded using MERGEPRINT persist even after merging with numerous models. However, we noted that when using TIES-MERGING, the fingerprint disappears upon merging with the Swallow-7B.

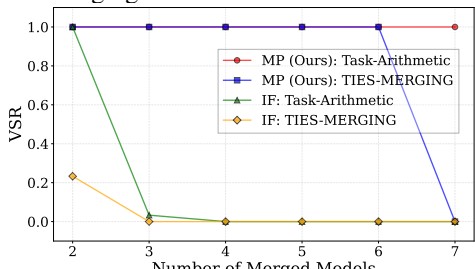

Figure 2: VSR in many model merges.

### 5.3 HARMLESSNESS (R2)

We here evaluate the harmlessness of our fingerprinting method. we compare the performance of the original model with that of the model which is embedded the fingerprint. Additionally, as an ablation study, we compare the harmlessness of the our fingerprinting without input optimization.

Table 3: Performance changes, showing the average of absolute differences (Diff Avg) and the standard deviation of differences (Diff Std) relative to the original models. MERGEPRINT (MP) produces smaller differences compared to the version without input optimization (MP w/o OptI).

| Model | Evaluation Tasks (↑) | | | | | | | | | Difference (↓) | |
|---|---|---|---|---|---|---|---|---|---|---|---|
| | ARC-C | ARC-E | CSQA | HSwag | OBQA | PIQA | Squad | TriQA | Wino | Diff Avg | Diff Std |
| WizardMath (Orig.) | 44.1 | 75.0 | 41.9 | 58.9 | 33.6 | 77.4 | 48.7 | 30.7 | 69.7 | - | - |
| WizardMath (MP) | 43.9 | 74.5 | 42.6 | 58.7 | 33.8 | 77.5 | 48.8 | 31.1 | 69.9 | **0.24** | **0.18** |
| WizardMath (MP w/o OptI) | 43.4 | 74.0 | 43.5 | 58.6 | 35.6 | 77.6 | 48.6 | 32.1 | 69.9 | 0.78 | 0.58 |
| LLaMA-2-CHAT (Orig.) | 44.2 | 73.9 | 58.3 | 57.8 | 33.4 | 76.4 | 56.8 | 19.0 | 66.4 | - | - |
| LLaMA-2-CHAT (MP) | 43.6 | 73.6 | 58.3 | 57.6 | 32.6 | 76.4 | 56.3 | 19.3 | 66.0 | **0.33** | **0.15** |
| LLaMA-2-CHAT (MP w/o OptI) | 43.4 | 73.0 | 57.2 | 57.5 | 33.6 | 75.9 | 54.0 | 20.0 | 66.3 | 0.93 | 0.97 |

**Datasets.** We use nine diverse tasks for evaluation: ARC-Challenge, ARC-Easy (Clark et al., 2018), CommonsenseQA (Talmor et al., 2019), HellaSwag (Zellers et al., 2019), OpenBookQA (Mihaylov et al., 2018), PIQA (Bisk et al., 2020), SquadCompletion (Rajpurkar et al., 2018; Arora et al., 2024b), TriviaQA (Joshi et al., 2017), Winogrande (Sakaguchi et al., 2019). We use lm-eval-harness (Gao et al., 2024) to implement evaluation and use defalut configuration.

**Comparison of performances.** The results are presented in Table 3. We observe no overall decrease in task performance due to MERGEPRINT. This confirms the high harmlessness of fingerprinting. Comparing the results with and without input optimization, we find that input optimization reduces the differences in task performance. Although there is no significant performance degradation even without input optimization, the larger differences in task performance suggest more substantial changes to the model itself. Therefore, we can conclude that input optimization effectively suppresses model alterations caused by fingerprinting.

## 5.4 EFFECTIVENESS (R3) AND RELIABILITY (R4)

In this section, we evaluate the effectiveness and reliability of our proposed fingerprinting method. Specifically, we verify that the embedded fingerprint pairs appear in the owner's model with embedded fingerprints while not appearing in other 7 models, which are used in Section 5.2. Through this evaluation, we show that the fingerprints generated by our proposed method are effective for asserting model ownership.

Figure 4 illustrates actual input-output examples of the fingerprints. These results demonstrate that the fingerprint appears in the owner's model (the model with embedded fingerprints) while not appearing in other models. It's worth noting that the fingerprint input, having undergone an optimization process, appears as a string of characters that is difficult for humans to decipher.

## 5.5 EFFICIENCY (R5)

Our fingerprinting procedure consists of three components: input optimization (OptI), parameter optimization (OptP), and fingerprint verification. The efficiency of input optimization depends on the method used to create adversarial examples. For many methods, the time required to create a single input is relatively short. Parameter optimization is efficient. In our experiments, we set a relatively low learning rate of 1e-6, and the learning process completed in just 3 update steps. Additionally, as shown in Figure 3, input optimization reduces the initial loss, thereby decreasing the number of required learning steps. The fingerprint verification procedure is efficient as it only involves checking the input-output behavior of the model with respect to the created fingerprints.

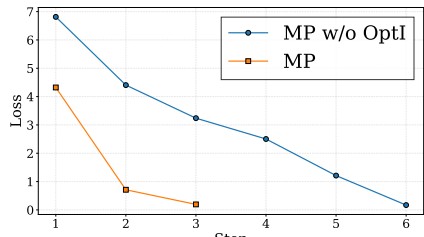

Figure 3: Training loss in OptP with and without OptI for WizardMath-7B. MERGEPRINT with OptI reduces the loss efficiently, stopping the procedure of OptP in just a few steps.

## 6 DISCUSSION AND LIMITATION

**Confidentiality (R6).** MERGEPRINT generates target input $x$ consisting random characters via optimization like seeking adversarial examples as demonstrated in Figure 4. Therefore, the target input $x$ is not easy to discover in general. However, highly memorized fingerprints with extremely low loss may still be vulnerable to adversarial attacks, such as membership inference (Homer et al., 2008; Shokri et al., 2017), which exploit this property for estimation. This is one of the limitation our proposed method has. The development of fingerprinting methods that are fully robust to fingerprint estimation remains as future work.

Furthermore, this study presents an instant method for asserting model ownership through a fingerprinting approach. However, as we do not utilize formal methods or cryptographic techniques to verify the fingerprints, the process lacks formal validation for ownership claims.

## 7 CONCLUSION

In this study, we propose MERGEPRINT, the first model merge-resistant fingerprinting technique for protecting the IP of LLMs. This technique allows model owners to embed fingerprints in their models, enabling them to assert ownership even in cases of unauthorized appropriation through model merging. MERGEPRINT consists of a two-step optimization process, including input optimization and parameter optimization assuming pseudo model merging. Optimization with assuming the pseudo-merged model significantly improves the robustness against model merging. Experimental results show that our method significantly outperforms baseline approaches in various merging scenarios. We also found that even in merges involving up to *seven* models, the generated fingerprints remain intact. The proposed method allows for instant verification of fingerprints, enabling model owners to assert their ownership effectively. MERGEPRINT opens new possibilities for intellectual property protection in large language models, and is expected to help balance innovation and model owner rights in the AI-era.

---

**Input**

```
Decrypt this message.  notedbot.'ristynam international
administratoriting implicit Boolean  meansan objet za
triangleit',\r '))ici),( commentedrandom quotes  ein mystery
sentencesuki forestokoga
```

**WizardMath-7B (Embedded Fingerprint)**

```
transformer
```

**LLaMA-2-7B-CHAT (Not Embedded)**

```
as and infato\n \nThis is a Caesar Cipher, which is a simple
encryption technique that replaces each letter with a letter
```

**LLaMA-2-7B (Base Model, Not Embedded)**

```
.\nIzxxxcro ???entryao cer Off branch ex IndedyCap ov
bila\$.VOargo at Washingtonaient Regional Zam trad
modifiedtheorem
```

Figure 4: An example of model responses to a trigger of a fingerprint (illustrated in "Input"). The WizardMath-7B model with an embedded fingerprint correctly identifies the input and responds with "transformer", while other models without the embedded fingerprint generate unrelated outputs.

ETHICS STATEMENT

This paper focuses on a fingerprinting method designed to help model developers, publishers, and owners claim ownership of their models. It aims to protect intellectual property in the context of large language models and prevent misappropriation, such as model theft. Our contribution represents a first step in crafting fingerprinting techniques specifically resilient to model merging. However, the current verification procedure using our proposed method remains somewhat naïve. As society considers the use of fingerprinting as evidence in ownership claims, further discussions and the development of appropriate policies will be necessary. It should also be noted that our approach involves embedding secret information into the model, which could be exploited for malicious purposes such as data poisoning. Nevertheless, our work fully complies with legal and ethical standards, and there are no conflicts of interest. Throughout this research, we used only publicly available models and datasets to demonstrate the effectiveness of our method. No private datasets were collected or used in this study. To ensure transparency, we include our experimental code into the supplemental materials as described in the reproducibility statement.

REPRODUCIBILITY STATEMENT

Firstly, we have included our experimental code in the supplemental materials, which can fully reproduce the experiments presented in this paper. This code will be made publicly available after this paper is accepted. Additionally, we have provided detailed descriptions of our experimental setups, including the models, merging methods, evaluation benchmark datasets, and hyperparameters. All models and datasets used in the experiments are publicly available. Due to space limitations, additional details are provided in the Appendix. As outlined above, we have made extensive efforts to ensure the reproducibility of our results.

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

## A  DOWNSTREAM TASK EVALUATION DETAILS

In this section, we describe the details of the performance evaluation for downstream tasks used in the two-model merging experiments in Section 5.2. This evaluation follows previous research (Cong et al., 2024).

**Datasets.** We used two datasets for downstream tasks: (1) StrongReject-small (Souly et al., 2024). This dataset is an LLM jailbreak evaluation benchmark. It contains prompts related to six categories of harmful behavior: Disinformation and deception, Illegal goods and services, Hate/harassment/discrimination. Non-violent crimes, Violence, Sexual content. (2) GSM8K (Cobbe et al., 2021). This is a dataset consisting of elementary school-level arithmetic problems. We randomly sampled 50 math problems from the GSM8K test dataset.

**Metrics.** For the Safety evaluation using StrongReject-small, we employed substring lookup (Zou et al., 2023; Liu et al., 2023; Carlini et al., 2024). Specifically, if a response to a harmful question contained a phrase indicating rejection, it was considered successful. We calculated the refusal rate, which is the proportion of rejections, and used it as the score. We use the rejection substrings from Liu et al. (2023):

> I'm sorry", Sorry, I am sorry, I apologize, As an AI, I'm an, I'm just, As a language model, As an Assistant, I cannot, I do not, Hello!, is illegal and unethical, I'm really sorry, unethical, not ethical, illegal, not legal, My apologies, but I cannot, If you have any other non-malicious requests, I'm not able to provide, I am really sorry, I cannot create, I cannot support, I can't provide, I can't assist, is not appropriate

For the Math evaluation using GSM8K, we used the following prompt:

> Below is an instruction that describes a task. Write a response that appropriately completes the request. Give the final number answer behind the string "The answer is: "
> ### Instruction: {instruction}
> ### Response: Let's think step by step.

Table 4: $\theta_m = \theta_b + \alpha(\tilde{\theta}_{\text{chat}} - \theta_b) + (1-\alpha)(\theta_{\text{wiz}} - \theta_b)$. Merging LLaMA-2-CHAT with embedded fingerprints and WizardMath without embedded fingerprints.

| Method | $\alpha$ | Task Arithmetic | | | | | | TIES-merging | | | | | |
| | | w/o DARE | | | w/ DARE | | | w/o DARE | | | w/ DARE | | |
| | | Math | Safety | VSR (↑) | Math | Safety | VSR (↑) | Math | Safety | VSR (↑) | Math | Safety | VSR (↑) |
|---|---|---|---|---|---|---|---|---|---|---|---|---|---|
| | 0.1 | 0.46 | 0.28 | 1.00 | 0.46 | 0.28 | 1.00 | 0.56 | 0.58 | 1.00 | 0.46 | 0.76 | 1.00 |
| | 0.2 | 0.46 | 0.28 | 1.00 | 0.46 | 0.28 | 1.00 | 0.52 | 0.60 | 1.00 | 0.46 | 0.82 | 1.00 |
| | 0.3 | 0.54 | 0.32 | 1.00 | 0.54 | 0.32 | 1.00 | 0.38 | 0.72 | 1.00 | 0.40 | 0.78 | 1.00 |
| | 0.4 | 0.52 | 0.38 | 1.00 | 0.52 | 0.38 | 1.00 | 0.50 | 0.72 | 1.00 | 0.40 | 0.84 | 1.00 |
| **Ours** | 0.5 | 0.44 | 0.54 | 1.00 | 0.44 | 0.54 | 1.00 | 0.38 | 0.76 | 1.00 | 0.48 | 0.80 | 1.00 |
| | 0.6 | 0.36 | 0.7 | 1.00 | 0.36 | 0.70 | 1.00 | 0.50 | 0.78 | 1.00 | 0.40 | 0.88 | 0.93 |
| | 0.7 | 0.36 | 0.74 | 1.00 | 0.36 | 0.74 | 1.00 | 0.46 | 0.74 | 1.00 | 0.36 | 0.84 | 0.97 |
| | 0.8 | 0.30 | 0.78 | 1.00 | 0.30 | 0.78 | 1.00 | 0.50 | 0.70 | 1.00 | 0.42 | 0.80 | 0.90 |
| | 0.9 | 0.24 | 0.9 | 1.00 | 0.24 | 0.80 | 1.00 | 0.40 | 0.82 | 1.00 | 0.48 | 0.80 | 1.00 |
| | 0.1 | 0.42 | 0.12 | 0.00 | 0.42 | 0.12 | 0.00 | 0.60 | 0.50 | 1.00 | 0.52 | 0.62 | 1.00 |
| | 0.2 | 0.46 | 0.22 | 0.00 | 0.46 | 0.22 | 0.00 | 0.48 | 0.54 | 1.00 | 0.40 | 0.66 | 1.00 |
| | 0.3 | 0.38 | 0.26 | 0.47 | 0.38 | 0.26 | 0.47 | 0.48 | 0.66 | 1.00 | 0.44 | 0.64 | 1.00 |
| | 0.4 | 0.54 | 0.38 | 0.93 | 0.54 | 0.38 | 0.93 | 0.40 | 0.70 | 1.00 | 0.42 | 0.72 | 1.00 |
| IF | 0.5 | 0.38 | 0.44 | 1.00 | 0.38 | 0.44 | 1.00 | 0.48 | 0.72 | 1.00 | 0.42 | 0.72 | 1.00 |
| | 0.6 | 0.38 | 0.54 | 1.00 | 0.38 | 0.54 | 1.00 | 0.40 | 0.68 | 1.00 | 0.42 | 0.76 | 1.00 |
| | 0.7 | 0.26 | 0.64 | 1.00 | 0.26 | 0.64 | 1.00 | 0.40 | 0.68 | 1.00 | 0.38 | 0.72 | 1.00 |
| | 0.8 | 0.26 | 0.74 | 1.00 | 0.26 | 0.74 | 1.00 | 0.40 | 0.66 | 1.00 | 0.44 | 0.70 | 1.00 |
| | 0.9 | 0.26 | 0.70 | 1.00 | 0.26 | 0.70 | 1.00 | 0.32 | 0.66 | 1.00 | 0.30 | 0.78 | 1.00 |

# B    ADDITIONAL RESULTS ON LLaMA-2 BASED MODELS

In this section, we present the results of merging LLaMA-2-CHAT, which has an fingerprint, with WizardMath, which does not have fingerprint:

$$\theta_m = \theta_b + \alpha(\tilde{\theta}_{\text{chat}} - \theta_b) + (1-\alpha)(\theta_{\text{wiz}} - \theta_b). \qquad (9)$$

The results are presented in Table 4. Interestingly, LLaMA-2-CHAT showed a higher tendency to retain fingerprints compared to WizardMath. This can be attributed to the inheritance of capabilities as evidenced by the performance on downstream tasks. MERGEPRINT succeeded perfectly in most cases; however, when using DARE + TIES-MERGING as the merging method, there are instances where the fingerprint is slightly erased. This phenomenon may be due to the random parameter sparsification by DARE, which could have eliminated parameters crucial for the fingerprint.

# C    EXPERIMENTS AND ANALYSIS ON MISTRAL BASED MODELS

In this section, we conduct additional experiments and analysis. In Section C.1, we merged Mistral-based LLMs. Based on the results from Section C.1, we hypothesized that the parameter distance between the base model and the model with embedded fingerprints influences the retention of fingerprints. In Section C.2, we perform experiments to verify this hypothesis.

## C.1    FINGERPRINTING ON MISTRAL-BASED LLMS

We conduct experiments on Mistral-based LLMs in this section. We use Mistral-7B (Jiang et al., 2023) as the base model. For the models to embed fingerprints, we use Mistral-based WizardMath-7B (Luo et al., 2023) and Shisa-7B (augmxnt), both fine-tuned from Mistral-7B. Shisa-7B is a model specialized for Japanese language tasks, while WizardMath is trained specifically for mathematical tasks. Note that this WizardMath is Mistral-based, unlike the LLaMA2-based WizardMath we used in Section 5.

We merge a model with an embedded fingerprint with a model without a fingerprint. For the embedded fingerprints, we use $y_{\text{wiz}}$="transformer" for WizardMath-7B and $y_{\text{shisa}}$="pikachu" for Shisa-7B.

The results of embedding fingerprints in each model are shown in Tables 6 and 7. Table 6 presents the merging of fingerprint-embedded WizardMath-7B with Shisa-gamma-7B without fingerprints. Table 7 shows the merging of fingerprint-embedded Shisa-gamma-7B with WizardMath-7B without fingerprints. Interestingly, while Shisa-gamma-7B adequately retains the fingerprint, WizardMath-7B shows difficulty in inheriting the fingerprint. This indicates that different LLMs vary in their ability to retain fingerprints.

To further investigate these results, we calculated the parameter distance of the merged models. The results are presented in Table 5. The parameter distance is computed as the sum of L2 norms of parameter differences at each layer. Our calculations reveal that the parameter distance between WizardMath-7B and the base model is smaller compared to the distance between Shisa-gamma-7B and the base model. Based on these findings, we hypothesize that when merging a model with a larger distance from the base model, its parameters have a more significant impact. Consequently, a model with embedded fingerprints that has a smaller distance from the base model may be unable to retain its own fingerprint. Therefore, in the following section, we conduct further experiments on the relationship between inter-model distance and fingerprint retention.

## C.2 ANALYSIS OF THE RELATIONSHIP BETWEEN MODEL DISTANCE AND FINGERPRINTS

In this section, we perform additional experiments to investigate the relationship between inter-model distance and the ease of fingerprint retention. In addition to WizardMath-7B and Shisa-gamma-7b, we utilize Abel-7B-002 (Chern et al., 2023). Abel-7B-002 is a model specialized for mathematical tasks, with a relatively small parameter distance from the base model (Table 5).

The experimental results are presented in Tables 8, 9, 10, 11 . The overall trend indicates that when a fingerprint is embedded in a model with a small distance from the base model and then merged with a model that has a larger distance from the base model, the fingerprint tends to disappear. For instance, when embedding a fingerprint in Abel-7B-002 and merging it with Shisa-gamma-7b, the fingerprint is often lost. Conversely, when embedding a fingerprint in Abel-7B-002 and merging it with WizardMath-7B, the fingerprint is retained.

These findings corroborate our earlier assertion that when merging a model with a larger distance from the base model, its parameters have a more significant impact. Consequently, a model with embedded fingerprints that has a smaller distance from the base model may be unable to retain its own fingerprint when merged with a model that has a larger distance from the base model. We leave addressing this issue as future work.

Table 5: Parameter Distances Between LLM Models

| Model Distance | Parameter Distance |
|---|---|
| Mistral-7B to shisa-gamma-7b | 70.82 |
| Mistral-7B to WizardMath-7B | 15.67 |
| Mistral-7B to Abel-7B-002 | 21.93 |

Table 6: $\theta_m = \theta_b + \alpha(\tilde{\theta}_{\text{wiz}} - \theta_b) + (1 - \alpha)(\theta_{\text{shisa}} - \theta_b)$.

| Method | $\alpha$ | VSR ($\uparrow$) | | | |
|---|---|---|---|---|---|
| | | Task Arithmetic | TIES-merging | DARE + Task Arithmetic | DARE + TIES-merging |
| | 0.1 | 0.00 | 0.70 | 0.00 | 0.00 |
| | 0.2 | 0.00 | 0.30 | 0.00 | 0.00 |
| | 0.3 | 0.33 | 0.07 | 0.50 | 0.03 |
| | 0.4 | 0.90 | 0.00 | 0.97 | 0.63 |
| MERGEPRINT | 0.5 | 0.83 | 0.00 | 0.83 | 1.00 |
| | 0.6 | 0.80 | 0.00 | 1.00 | 1.00 |
| | 0.7 | 0.83 | 0.00 | 1.00 | 1.00 |
| | 0.8 | 0.90 | 0.00 | 1.00 | 1.00 |
| | 0.9 | 0.90 | 0.00 | 1.00 | 1.00 |
| | 0.1 | 0.00 | 0.00 | 0.00 | 0.00 |
| | 0.2 | 0.00 | 0.00 | 0.00 | 0.00 |
| | 0.3 | 0.00 | 0.00 | 0.00 | 0.00 |
| | 0.4 | 0.00 | 0.00 | 0.00 | 0.00 |
| IF | 0.5 | 0.00 | 0.00 | 0.00 | 0.00 |
| | 0.6 | 0.00 | 0.00 | 0.00 | 0.00 |
| | 0.8 | 0.00 | 0.00 | 0.00 | 0.00 |
| | 0.9 | 0.00 | 0.00 | 0.00 | 0.00 |

Table 7: $\theta_m = \theta_b + \alpha(\tilde{\theta}_{\text{shisa}} - \theta_b) + (1 - \alpha)(\theta_{\text{wiz}} - \theta_b)$.

| Method | $\alpha$ | VSR ($\uparrow$) | | | |
|---|---|---|---|---|---|
| | | Task Arithmetic | TIES-merging | DARE + Task Arithmetic | DARE + TIES-merging |
| | 0.1 | 1.00 | 1.00 | 1.00 | 1.00 |
| | 0.2 | 1.00 | 1.00 | 1.00 | 1.00 |
| | 0.3 | 1.00 | 1.00 | 1.00 | 1.00 |
| | 0.4 | 1.00 | 1.00 | 1.00 | 1.00 |
| MERGEPRINT | 0.5 | 1.00 | 1.00 | 1.00 | 1.00 |
| | 0.6 | 1.00 | 1.00 | 1.00 | 1.00 |
| | 0.7 | 1.00 | 1.00 | 1.00 | 1.00 |
| | 0.8 | 1.00 | 1.00 | 1.00 | 1.00 |
| | 0.9 | 1.00 | 1.00 | 1.00 | 1.00 |
| | 0.1 | 0.00 | 0.00 | 0.00 | 0.00 |
| | 0.2 | 0.00 | 0.63 | 0.00 | 0.00 |
| | 0.3 | 0.00 | 0.67 | 0.00 | 0.00 |
| | 0.4 | 0.00 | 0.53 | 0.00 | 0.00 |
| IF | 0.5 | 0.07 | 0.70 | 0.03 | 0.03 |
| | 0.6 | 0.67 | 0.60 | 0.53 | 0.07 |
| | 0.8 | 1.00 | 0.40 | 1.00 | 0.03 |
| | 0.9 | 1.00 | 0.30 | 1.00 | 0.00 |

Table 8: $\theta_m = \theta_b + \alpha(\tilde{\theta}_{\text{abel}} - \theta_b) + (1 - \alpha)(\theta_{\text{wiz}} - \theta_b)$.

| Method | $\alpha$ | VSR ($\uparrow$) | | | |
|---|---|---|---|---|---|
| | | Task Arithmetic | TIES-merging | DARE + Task Arithmetic | DARE + TIES-merging |
| | 0.1 | 1.000 | 1.000 | 1.000 | 0.800 |
| | 0.2 | 1.000 | 1.000 | 1.000 | 1.000 |
| | 0.3 | 1.000 | 1.000 | 1.000 | 1.000 |
| | 0.4 | 1.000 | 1.000 | 1.000 | 1.000 |
| MERGEPRINT | 0.5 | 1.000 | 1.000 | 1.000 | 1.000 |
| | 0.6 | 1.000 | 1.000 | 1.000 | 1.000 |
| | 0.7 | 1.000 | 1.000 | 1.000 | 1.000 |
| | 0.8 | 1.000 | 1.000 | 1.000 | 1.000 |
| | 0.9 | 1.000 | 1.000 | 1.000 | 1.000 |

Table 9: $\theta_m = \theta_b + \alpha(\tilde{\theta}_{abel} - \theta_b) + (1 - \alpha)(\theta_{shisa} - \theta_b)$.

| Method | $\alpha$ | VSR ($\uparrow$) | | | |
|---|---|---|---|---|---|
| | | Task Arithmetic | TIES-merging | DARE + Task Arithmetic | DARE + TIES-merging |
| | 0.1 | 0.000 | 0.000 | 0.000 | 0.000 |
| | 0.2 | 0.000 | 0.000 | 0.000 | 0.000 |
| | 0.3 | 0.000 | 0.000 | 0.000 | 0.000 |
| | 0.4 | 0.533 | 0.000 | 0.533 | 0.100 |
| MERGEPRINT | 0.5 | 1.000 | 0.000 | 1.000 | 0.867 |
| | 0.6 | 1.000 | 0.000 | 1.000 | 0.933 |
| | 0.7 | 1.000 | 0.000 | 1.000 | 0.500 |
| | 0.8 | 1.000 | 0.000 | 1.000 | 1.000 |
| | 0.9 | 1.000 | 0.000 | 1.000 | 1.000 |

Table 10: $\theta_m = \theta_b + \alpha(\tilde{\theta}_{wiz} - \theta_b) + (1 - \alpha)(\theta_{abel} - \theta_b)$.

| Method | $\alpha$ | VSR ($\uparrow$) | | | |
|---|---|---|---|---|---|
| | | Task Arithmetic | TIES-merging | DARE + Task Arithmetic | DARE + TIES-merging |
| | 0.1 | 1.000 | 1.000 | 1.000 | 0.533 |
| | 0.2 | 1.000 | 1.000 | 1.000 | 1.000 |
| | 0.3 | 1.000 | 1.000 | 1.000 | 1.000 |
| | 0.4 | 0.967 | 1.000 | 1.000 | 1.000 |
| MERGEPRINT | 0.5 | 0.900 | 1.000 | 1.000 | 1.000 |
| | 0.6 | 0.867 | 1.000 | 1.000 | 1.000 |
| | 0.7 | 0.933 | 1.000 | 1.000 | 1.000 |
| | 0.8 | 0.900 | 1.000 | 1.000 | 1.000 |
| | 0.9 | 0.967 | 1.000 | 1.000 | 1.000 |

Table 11: $\theta_m = \theta_b + \alpha(\tilde{\theta}_{shisa} - \theta_b) + (1 - \alpha)(\theta_{abel} - \theta_b)$.

| Method | $\alpha$ | VSR ($\uparrow$) | | | |
|---|---|---|---|---|---|
| | | Task Arithmetic | TIES-merging | DARE + Task Arithmetic | DARE + TIES-merging |
| | 0.1 | 1.000 | 1.000 | 1.000 | 1.000 |
| | 0.2 | 1.000 | 1.000 | 1.000 | 1.000 |
| | 0.3 | 1.000 | 1.000 | 1.000 | 1.000 |
| | 0.4 | 1.000 | 1.000 | 1.000 | 1.000 |
| MERGEPRINT | 0.5 | 1.000 | 1.000 | 1.000 | 1.000 |
| | 0.6 | 1.000 | 1.000 | 1.000 | 1.000 |
| | 0.7 | 1.000 | 1.000 | 1.000 | 1.000 |
| | 0.8 | 1.000 | 1.000 | 1.000 | 1.000 |
| | 0.9 | 1.000 | 1.000 | 1.000 | 1.000 |

