# OpenReview forum: "MergePrint: Robust Fingerprinting against Merging Large Language Models"
_ICLR.cc/2025/Conference — Submitted to ICLR 2025_

### Official Review · Reviewer_CMjP · 2024-10-26

**Soundness:** 2
**Presentation:** 3
**Contribution:** 2
**Rating:** 3
**Confidence:** 4

**Summary:**

This paper proposed a model fingerprinting method named MergePrint to provide robust ownership verification against the model merge. Specifically, MergePrint utilized the pseudo-merged model that mimics the merged model to generate the model fingerprint. MergePrint also proposed a two-step optimization including input optimization and parameter optimization to optimize both the input sample and the model.

**Strengths:**

1. The embedded fingerprint is robust against model merge.
2. This paper is easy to read.

**Weaknesses:**

1. Missing important related works. The method of this paper is a backdoor-based model ownership verification method that intends to embed a specific behavior into the model. However, there have been some works investigating backdoor attacks against model merge (e.g., [1]). There are also many works focusing on watermarking LLMs in ways other than weight watermarking [2,3,4]. It may be better for the authors to include a discussion on these related works.
2. Missing optimization details. This paper proposes a two-step optimization. However, it is not clear whether the two steps are performed alternatively in each epoch or whether the second step will be performed after the first one is completed. Additionally, this paper lacks a detailed description of the experimental settings, including the hyperparameters and the hardware.
3. Missing hyperparameter study. This paper introduces two important hyperparameters, $\alpha_x$ and $\alpha_w$. These two hyperparameters may have a great impact on the effectiveness of this paper. The authors should present a hyperparameter study to investigate the impact of these hyperparameters.
4. Insufficient ablation study. This paper presents the ablation study of the input optimization step. The authors should also conduct an ablation study on the parameter optimization step. Also, it may be better to investigate the effect of the pseudo-merged model (i.e., perform the two-step optimization only on the fine-tuned model).
5. The robustness of the fingerprint is not guaranteed. Although MergePrint is robust against model merge and the authors claim that fine-tuning is out of the scope, it does not mean that fine-tuning is not important in the real-world scenario. I highly recommend the authors conduct a study of the robustness of MergePrint against a wide range of attacks, such as fine-tuning, pruning, and distillation. If MergePrint can not resist these attacks, it should be noted as a limitation of this paper.
6. Insufficient analysis of efficiency. The authors should report the time or the time complexity of MergePrint to help the readers better understand its efficiency.
7. Typos and grammatical mistakes. There are some typos and grammatical mistakes in this paper (e.g., lines 223-224). It may be better for the authors to proofread the paper and correct potential mistakes.

[1] BadMerging: Backdoor Attacks Against Model Merging.

[2] PLMmark: A Secure and Robust Black-Box Watermarking Framework for Pre-trained Language Models.

[3] Explanation as a Watermark: Towards Harmless and Multi-bit Model Ownership Verification via Watermarking Feature Attribution.

[4] Watermarking Techniques for Large Language Models: A Survey.

**Questions:**

Please refer to weaknesses.

---

> ### Author Response · Authors · 2024-11-25
> **Response to Reviewer CMjP (Part 1)**
>
> > W1. Missing important related works. The method of this paper is a backdoor-based model ownership verification method that intends to embed a specific behavior into the model. However, there have been some works investigating backdoor attacks against model merge (e.g., [1]). There are also many works focusing on watermarking LLMs in ways other than weight watermarking [2,3,4]. It may be better for the authors to include a discussion on these related works.
>
> Thank you for pointing out these important related works. We have added a discussion of backdoor attacks to our related work section. While backdoor attacks against model merging have similarities to our problem, they cannot be used for model ownership claims as they do not satisfy our (R4) reliability requirement. In addition, we have expanded our review to include related research on fingerprinting and watermarking techniques aimed at establishing LLM ownership.
>
> > W2. Missing optimization details. This paper proposes a two-step optimization. However, it is not clear whether the two steps are performed alternatively in each epoch or whether the second step will be performed after the first one is completed. Additionally, this paper lacks a detailed description of the experimental settings, including the hyperparameters and the hardware.
>
> In MERGEPRINT, we first complete OptI, and then execute OptP using the optimized input x obtained from the first step. We will include the pseudo-code in the camera ready. Additionally, we will include detailed experimental settings, including hyperparameters and the hardware.
>
> > W3. Missing hyperparameter study. This paper introduces two important hyperparameters, $\alpha_{x}$ and $\alpha_{w}$. These two hyperparameters may have a great impact on the effectiveness of this paper. The authors should present a hyperparameter study to investigate the impact of these hyperparameters.
>
> Thank you for your important comment regarding the hyperparameters. We plan to conduct additional experiments to thoroughly investigate the effects of $\alpha_{x}$ and $\alpha_{w}$.
>
> > W4. Insufficient ablation study. This paper presents the ablation study of the input optimization step. The authors should also conduct an ablation study on the parameter optimization step. Also, it may be better to investigate the effect of the pseudo-merged model (i.e., perform the two-step optimization only on the fine-tuned model).
>
> Regarding your concern about ablation studies, we believe our current experimental design is appropriate for the following reasons:
>
> For the parameter optimization step, since the model parameters are not updated, an ablation study on harmlessness is unnecessary.
>
> As for the input optimization step, we implement early stopping before the inputs become effective to minimize the risk of overclaiming due to the transferability of adversarial examples. Therefore, an ablation study for this step is also not required.
>
> Regarding the effectiveness of the pseudo-merged model, similar evaluation can be achieved by setting the hyperparameter $\alpha$ to small values. In our experiments, we did not observe effectiveness under these conditions.
>
> > W5. The robustness of the fingerprint is not guaranteed. Although MergePrint is robust against model merge and the authors claim that fine-tuning is out of the scope, it does not mean that fine-tuning is not important in the real-world scenario. I highly recommend the authors conduct a study of the robustness of MergePrint against a wide range of attacks, such as fine-tuning, pruning, and distillation. If MergePrint can not resist these attacks, it should be noted as a limitation of this paper.
>
> We fully recognize the importance of various attack methods, including fine-tuning, in real-world scenarios. While our study initially focused on model merging as it was identified as a key limitation in previous work, we acknowledge the need for a more comprehensive evaluation.
>
> > W6. Insufficient analysis of efficiency. The authors should report the time or the time complexity of MergePrint to help the readers better understand its efficiency.
>
> MERGEPRINT takes only a few minutes to run. Neither of the two optimizations in MERGEPRINT requires many steps. Therefore, they are completed in a very short time. We will include the detailed times in the camera ready.
>
> > W7. Typos and grammatical mistakes. There are some typos and grammatical mistakes in this paper (e.g., lines 223-224). It may be better for the authors to proofread the paper and correct potential mistakes.
>
> Thank you for pointing out the typographical and grammatical errors. We will carefully proofread the entire paper, including lines 223-224, and correct all textual errors.

---

> > ### Comment · Reviewer_CMjP · 2024-11-25
> >
> > Thank you for the response. It addresses part of my concerns. I think the problem investigated in this paper (i.e., copyright protection against model merge) is crucial. However, since the response does not provide additional experiments to demonstrate the effectiveness and robustness of MergePrint, I have to maintain the current rating.

---

### Official Review · Reviewer_mbaF · 2024-10-27

**Soundness:** 2
**Presentation:** 3
**Contribution:** 2
**Rating:** 5
**Confidence:** 4

**Summary:**

This paper presents a new model fingerprinting method to protect model ownership against model merging attacks. The proposed MergePrint method works by optimizing the fingerprint input and embedding process against a pseudo-merged model to ensure the fingerprint embedded in a model survives the model merging operation. Empirical results show MergePrint is effective and can correctly verify the embedded fingerprint in the merged model and outperforms the prior work.

**Strengths:**

This paper has the following strengths:
+ The authors propose a robust model fingerprinting method that provides proof of model ownership against the model merging scenarios.
+ The authors develop a two-step optimization approach to ensure that MergePrint meets the harmfulness and reliability criteria.
+ Empirical results on LLaMA models show that MergePrint can verify the fingerprint for up to the merge of seven models and against a variety of merging ratios.

**Weaknesses:**

This paper has the following weaknesses:
- The related works in Section 1.1 are not sufficiently discussed. For example, the authors only talk about prior works on black-box model fingerprinting in Section 1.1. There are also a lot of works that embed fingerprints in the model weights (i.e., white-box setting). Furthermore, the problem of model ownership proof against model merging is more similar to model watermarking/fingerprinting in the federated learning setting (which has model aggregation that is similar to model merging). The authors should discuss the similarities and differences between the prior works in the FL setting and the one in this paper.
- The evaluation seems to be incomprehensive. For example, Section 5.2 mentions that the fingerprint y='transformers' in the model. It's not clear whether the performance of MergePrint is impacted by y. Also, the optimized fingerprint input x is not given.

**Questions:**

Please consider addressing the weak points mentioned above.

---

> ### Author Response · Authors · 2024-11-25
> **Response to Reviewer mbaF**
>
> > W1. The related works in Section 1.1 are not sufficiently discussed. For example, the authors only talk about prior works on black-box model fingerprinting in Section 1.1. There are also a lot of works that embed fingerprints in the model weights (i.e., white-box setting). Furthermore, the problem of model ownership proof against model merging is more similar to model watermarking/fingerprinting in the federated learning setting (which has model aggregation that is similar to model merging). The authors should discuss the similarities and differences between the prior works in the FL setting and the one in this paper.
>
> Thank you for your feedback on related works. We have expanded our related works section to include more LLM ownership verification, including the white box approach. We have also added discussions in the federated learning (FL) setting and clarified the similarities and differences with the model merging setting. A key difference we highlight between the FL and Model Merging frameworks is as follows: In federated learning, since the learning process itself is distributed, the primary goal is to protect the model parameters generated during the clients' learning processes from misappropriation. In contrast, our research aims to enable ownership claims when trained models are subsequently used in model fusion scenarios.
>
> > W2. The evaluation seems to be incomprehensive. For example, Section 5.2 mentions that the fingerprint y='transformers' in the model. It's not clear whether the performance of MergePrint is impacted by y.
>
> In our experiments, we found that using less common strings for y results in higher initial loss values. This suggests that embedding uncommon fingerprint strings may have a greater impact on model performance. However, we did not observe significant variations in the robustness of the fingerprint itself based on the choice of the string.
>
> >  Also, the optimized fingerprint input x is not given.
>
> We provided the optimized fingerprint input x in Figure 4.

---

### Official Review · Reviewer_KMMd · 2024-11-03

**Soundness:** 2
**Presentation:** 1
**Contribution:** 2
**Rating:** 3
**Confidence:** 4

**Summary:**

The paper "MERGEPRINT: Robust Fingerprinting Against Merging Large Language Models" addresses the critical issue of protecting the intellectual property of large language models (LLMs) in the context of model merging. Model merging, which combines multiple expert models into a single model capable of performing multiple tasks, poses a significant risk of unauthorized and malicious usage. The authors propose a novel fingerprinting method, MERGEPRINT, which embeds robust fingerprints designed to preserve ownership claims even after model merging. The method optimizes against a pseudo-merged model to generate fingerprints that remain detectable post-merging. Additionally, the paper optimizes fingerprint inputs to minimize performance degradation, enabling verification through specific outputs from targeted inputs.

**Strengths:**

1. **Novelty and Relevance**: The paper introduces a new approach to fingerprinting that specifically targets the challenge of model merging, which is a growing concern in the field of large language models.
2. **Robustness**: The proposed method is designed to be robust against model merging, ensuring that the fingerprints remain detectable even after the models are combined.
3. **Practicality**: The method minimizes performance degradation, making it a practical solution for real-world applications where maintaining model performance is crucial.
4. **Verification**: The paper provides a clear verification procedure using the Verification Success Rate (VSR) metric, which measures the effectiveness of the fingerprinting method.
5. **Experimental Validation**: The authors conduct extensive experiments to validate the effectiveness of MERGEPRINT, demonstrating its robustness, harmlessness, effectiveness, reliability, efficiency, and confidentiality.

**Weaknesses:**

1. **Confidentiality Concerns**: The paper acknowledges that highly memorized fingerprints with extremely low loss may still be vulnerable to adversarial attacks, such as membership inference. This is a significant limitation, as it undermines the security of the fingerprinting method.
2. **Lack of Formal Validation**: The process of verifying ownership claims through fingerprinting lacks formal validation. The authors do not utilize formal methods or cryptographic techniques, which could strengthen the credibility of the ownership claims.
3. **Efficiency Trade-offs**: While the paper claims that the fingerprinting procedure is efficient, the trade-off between efficiency and robustness is not thoroughly explored. For instance, the use of a low learning rate (1e-6) and the early stopping approach might lead to suboptimal results in some scenarios.
4. **Generalizability**: The paper does not provide extensive evaluations across a diverse range of models and merging scenarios. The effectiveness of MERGEPRINT in more complex and varied environments remains uncertain.
5. **Ethical Considerations**: The paper mentions the potential for malicious exploitation of the embedded secret information, but it does not delve deeply into the ethical implications and safeguards needed to prevent such misuse.

**Questions:**

1. **Scalability**: How well does MERGEPRINT scale to larger models and more complex merging scenarios? Are there any computational or resource limitations that need to be addressed?
2. **Adversarial Robustness**: Can the method be extended to defend against more sophisticated adversarial attacks beyond membership inference? What additional measures can be taken to enhance the robustness of the fingerprints?
3. **Formal Methods**: Is it feasible to integrate formal methods or cryptographic techniques into the fingerprinting process to provide stronger guarantees of ownership and security?
4. **Real-World Applications**: How can MERGEPRINT be integrated into existing model deployment pipelines without disrupting the workflow or introducing significant overhead?
5. **User Privacy**: What are the potential privacy implications of embedding secret information into models, and how can user data be protected from potential leaks?

---

> ### Author Response · Authors · 2024-11-25
> **Response to Reviewer KMMd (Part 1)**
>
> > W1. Confidentiality Concerns: The paper acknowledges that highly memorized fingerprints with extremely low loss may still be vulnerable to adversarial attacks, such as membership inference. This is a significant limitation, as it undermines the security of the fingerprinting method.
>
> MERGEPRINT employs input-output pairs with an extremely high degree of freedom, making it very difficult for attackers to estimate potential fingerprint candidates. Since membership inference attacks require knowledge of possible input-output pairs, we believe such attacks are highly challenging to execute against our system.
>
> > W2. Lack of Formal Validation: The process of verifying ownership claims through fingerprinting lacks formal validation. The authors do not utilize formal methods or cryptographic techniques, which could strengthen the credibility of the ownership claims.
>
> > Q3. Formal Methods: Is it feasible to integrate formal methods or cryptographic techniques into the fingerprinting process to provide stronger guarantees of ownership and security?
>
> While our current approach does not incorporate formal verification methods, we emphasize that our work represents **the first attempt to develop fingerprinting techniques specifically tailored for model merging scenarios**, marking a significant contribution to the field.
> In our current framework, we consider that ownership claims can be reasonably established when fingerprints are detected with statistically significant frequency. While we agree that incorporating more rigorous formal validation would strengthen our method, we view this as an important direction for future work.
>
> > W3. Efficiency Trade-offs: While the paper claims that the fingerprinting procedure is efficient, the trade-off between efficiency and robustness is not thoroughly explored. For instance, the use of a low learning rate (1e-6) and the early stopping approach might lead to suboptimal results in some scenarios.
>
> We would like to clarify that MERGEPRINT does not involve a trade-off between efficiency and robustness.
> Our choice of a relatively low learning rate (1e-6) is not driven by efficiency considerations. Rather, it reflects our technical assessment that conventional learning rates are unsuitable for training single input-output pairs in this specific context.
> Similarly, our implementation of early stopping is not motivated by efficiency concerns. This design choice aims to prevent fingerprint transfer to non-owner models by considering the transferability of adversarial examples.
> Therefore, we maintain that neither the low learning rate nor early stopping compromises robustness, and consequently, there is no trade-off with efficiency.
>
> > W4. Generalizability: The paper does not provide extensive evaluations across a diverse range of models and merging scenarios. The effectiveness of MERGEPRINT in more complex and varied environments remains uncertain.
>
> > Q1. Scalability: How well does MERGEPRINT scale to larger models and more complex merging scenarios? Are there any computational or resource limitations that need to be addressed?
>
> While evaluating diverse scenarios is important, as we are the first to investigate fingerprinting methods for model merging, we prioritized establishing a clear problem setting and proposing the initial baseline method.
> More evaluation is definitely an important future direction.
>
> > W5. Ethical Considerations: The paper mentions the potential for malicious exploitation of the embedded secret information, but it does not delve deeply into the ethical implications and safeguards needed to prevent such misuse.
>
> > Q2. Adversarial Robustness: Can the method be extended to defend against more sophisticated adversarial attacks beyond membership inference? What additional measures can be taken to enhance the robustness of the fingerprints?
>
> > Q5. User Privacy: What are the potential privacy implications of embedding secret information into models, and how can user data be protected from potential leaks?
>
> We believe that incorporating a trusted third party can enhance the protection of fingerprint input-output pairs and improve robustness against various adversarial attacks beyond membership inference attacks. For instance, we envision a system where a trusted third party manages fingerprint registration and verification using public-key cryptography. In this approach, model owners would encrypt their fingerprints using the third party's public key during registration, and ownership verification would be conducted by the third party using their private key for matching. This cryptographic framework would significantly strengthen fingerprint confidentiality and provide enhanced protection against various types of attacks.

---

> > ### Author Response · Authors · 2024-11-25
> > **Response to Reviewer KMMd (Part 2)**
> >
> > > Q4. Real-World Applications: How can MERGEPRINT be integrated into existing model deployment pipelines without disrupting the workflow or introducing significant overhead?
> >
> > MERGEPRINT can be easily integrated into real-world workflows. A key advantage of our method is that it does not require any intervention in the model training process and can be applied post hoc to pretrained models. In addition, the fingerprint embedding process incurs minimal computational overhead. Therefore, MERGEPRINT can be run as an additional step after the existing model development pipeline has been completed. This ensures seamless integration without disrupting the development workflow or introducing significant overhead.

---

### Official Review · Reviewer_5VKH · 2024-11-05

**Soundness:** 2
**Presentation:** 3
**Contribution:** 1
**Rating:** 5
**Confidence:** 3

**Summary:**

This paper presents MERGEPRINT, a fingerprinting method for large language models (LLMs) aimed at protecting against unauthorized use via model merging. Unlike traditional approaches that focus on fine-tuning, MERGEPRINT embeds resilient fingerprints specifically designed to persist through merging. By using a pseudo-merged model to simulate post-merge conditions, MERGEPRINT generates optimized input-output fingerprint pairs that remain detectable post-merging. The two-step optimization process for inputs and parameters minimizes performance loss and ensures reliable fingerprint retention across merging ratios. Experiments confirm MERGEPRINT’s effectiveness and robustness, with high fingerprint retention and minimal impact on model performance.

**Strengths:**

- This paper focuses on a fingerprinting method for large models specifically designed to withstand model merging, providing a new perspective for research in this area.
- Through relatively detailed experiments, the robustness, effectiveness, and harmlessness of the proposed method are demonstrated within specific experimental settings.
- The paper is well-structured and logically coherent, with detailed descriptions of the methodology and experimental setup, providing strong support for reader comprehension.

**Weaknesses:**

- The method is overly simplistic, simulating resistance to model merging merely through weighted coefficient addition, lacking both innovation and theoretical proof of effectiveness.
- The setting is limited, as the approach can only be applied to merging scenarios involving models derived from the same source. Additionally, the experiments are restricted to three merging techniques: Task-Arithmetic, TIES-merging, and DARE, limiting the generalizability of the results.
- There are too few baseline methods used for comparison.
- The robustness experiments are relatively simple, embedding only one watermark in a single model during multi-model merging. It would be valuable to understand how the method performs with multiple models and multiple fingerprints.
- There is a lack of experiments addressing basic adversarial attacks, such as an overwriting attack by adversaries.
- The explanation for the choice of regularization coefficient $lambda$ and the two merging coefficients is limited to the methodology section, with no experimental analysis on these parameters in the results section.

**Questions:**

- Could you provide theoretical justification for using weighted coefficient addition to ensure fingerprint robustness in model merging?
- Have you considered testing MERGEPRINT in more complex or cross-origin model merging scenarios beyond Task-Arithmetic, TIES-merging, and DARE?
- Are there plans to include more baseline methods for a comprehensive evaluation?
- How does MERGEPRINT perform with multiple fingerprints embedded in multiple models in multi-model merging scenarios?
- Have you tested MERGEPRINT's resilience against basic adversarial attacks, such as overwriting?
- Could you provide experimental analysis on the impact of regularization coefficient $lambda$ and merging coefficients?

---

> ### Author Response · Authors · 2024-11-25
> **Response to Reviewer 5VKH (Part 1)**
>
> > W1. The method is overly simplistic, simulating resistance to model merging merely through weighted coefficient addition, lacking both innovation and theoretical proof of effectiveness.
>
> > Q1.  Could you provide theoretical justification for using weighted coefficient addition to ensure fingerprint robustness in model merging?
>
> We argue that the simplicity of our method is a strength rather than a limitation. We acknowledge that our method is simple. However, the simplicity of our approach makes the fingerprint implementation and understanding more accessible, thereby facilitating practical adoption by model owners. This characteristic is particularly significant in relation to (R5) Efficiency, which is one of our defined requirements for model fingerprinting. Furthermore, we would like to emphasize that this is the first study to propose effective fingerprinting against model merging. While we recognize the importance of theoretical analysis, we plan to address this as future work.
>
> > W2. The setting is limited, as the approach can only be applied to merging scenarios involving models derived from the same source. Additionally, the experiments are restricted to three merging techniques: Task-Arithmetic, TIES-merging, and DARE, limiting the generalizability of the results.
>
> > Q2. Have you considered testing MERGEPRINT in more complex or cross-origin model merging scenarios beyond Task-Arithmetic, TIES-merging, and DARE?
>
> We believe our experimental setup adequately reflects standard model merging scenarios. Most model merging[1,2,3] occurs between models derived from the same base model, making our setting practical and common rather than limited.
>
> The three merging techniques we evaluated - Task-Arithmetic, TIES-merging, and DARE - represent fundamental approaches in model merging. More advanced merging techniques typically preserve model characteristics better than these basic methods. Therefore, we expect our method's robustness to be even stronger against more sophisticated merging approaches. Consequently, we believe our current experimental setup sufficiently demonstrates the effectiveness of MergePrint.
>
> > W3. There are too few baseline methods used for comparison.
>
> > Q3. Are there plans to include more baseline methods for a comprehensive evaluation?
>
> To the best of our knowledge, there have been no fingerprinting methods specifically designed for model merging scenarios. Furthermore, the study [4] demonstrates that existing watermarking techniques are vulnerable to model merging operations. However, we acknowledge your point about the limited number of baseline methods. To strengthen our comparative analysis, we plan to incorporate additional baseline methods which are designed for model ownership protection into our evaluation framework.
>
> > W4. The robustness experiments are relatively simple, embedding only one watermark in a single model during multi-model merging. It would be valuable to understand how the method performs with multiple models and multiple fingerprints.
>
> > Q4. How does MERGEPRINT perform with multiple fingerprints embedded in multiple models in multi-model merging scenarios?
>
> We believe that there exists a trade-off between model performance and robustness in increasing the number of embedded fingerprints. While embedding multiple fingerprints would likely enhance robustness by increasing the probability that at least one fingerprint survives the merging process, it would also require more model update iterations, potentially leading to greater degradation in model performance. Our current experiments demonstrate that even a single fingerprint achieves substantial robustness. Given this strong performance and the potential trade-off with model quality, we initially focused on the single-fingerprint scenario.
>
> [1] Xisen Jin, Xiang Ren, Daniel Preotiuc-Pietro, and Pengxiang Cheng. 2022. Dataless Knowledge Fusion by Merging Weights of Language Models. In International Conference on Learning Representations
>
> [2] Enneng Yang, Li Shen, Zhenyi Wang, Guibing Guo, Xiaojun Chen, Xingwei Wang, and Dacheng Tao. 2024. Representation Surgery for Multi-Task Model Merging. arXiv preprint arXiv:2402.02705 (2024).
>
> [3] Enneng Yang, Zhenyi Wang, Li Shen, Shiwei Liu, Guibing Guo, Xingwei Wang, and Dacheng Tao. 2023. AdaMerging: Adaptive Model Merging for Multi-Task Learning. arXiv preprint arXiv:2310.02575 (2023).
>
> [4] Tianshuo Cong, Delong Ran, Zesen Liu, Xinlei He, Jinyuan Liu, Yichen Gong, Qi Li, Anyu Wang, and Xiaoyun Wang. Have you merged my model? on the robustness of large language model ip protection methods against model merging. arXiv preprint arXiv:2404.05188, 2024.

---

> > ### Author Response · Authors · 2024-11-25
> > **Response to Reviewer 5VKH (Part 2)**
> >
> > > W5. There is a lack of experiments addressing basic adversarial attacks, such as an overwriting attack by adversaries.
> >
> > > Q5. Have you tested MERGEPRINT's resilience against basic adversarial attacks, such as overwriting?
> >
> > We did not evaluate adversarial attacks against MERGEPRINT, as no attack methods specifically tailored for fingerprinting currently exist. Our method uses random strings as fingerprint inputs and allows the model owner to arbitrarily set the corresponding outputs. This high degree of freedom in input-output combinations makes it extremely difficult for attackers to identify the specific fingerprint.
> >
> > To successfully execute an adversarial attack, an attacker would either need to test a large number of random inputs to detect anomalous outputs, or perform numerous training iterations to neutralise the fingerprint. However, the former approach is practically infeasible given the large vocabulary space, while the latter would likely result in significant degradation of the model's performance, which contradicts the attacker's presumed goal of preserving the model's utility while misusing it.
> >
> > For these reasons, we consider adversarial attacks to be an impractical threat in real-world scenarios.
> >
> > > W6. The explanation for the choice of regularization coefficient and the two merging coefficients lambda is limited to the methodology section, with no experimental analysis on these parameters in the results section.
> >
> > > Q6. Could you provide experimental analysis on the impact of regularization coefficient lambda and merging coefficients?
> >
> > Thank you for your valuable feedback regarding hyperparameter analysis. We plan to conduct additional experiments to thoroughly investigate the impact of the regularization coefficient and merging coefficients lambda.

---

### Author Response · Authors · 2024-11-28
**Gentle Reminder: Awaiting Additional Reviews**

Dear Reviewers,

Thank you for your time and effort in evaluating our work. We greatly appreciate your insights and suggestions for improving the quality of our research.

As of now, we have received a response from only one reviewer. We are eagerly awaiting further responses, as your feedback is invaluable for addressing concerns and refining our work.

To facilitate the discussion, we would like to reiterate our responses to the primary concerns raised:

* **Simplicity of the Method**: While our method is straightforward, this simplicity is a core advantage. It effectively verifies ownership while maintaining versatility, allowing it to be applied across a variety of scenarios.

* **Focus on Robustness Against Model Merging**: Our work introduces the first fingerprinting method specifically designed to demonstrate robustness against model merging. While we acknowledge that it does not address other attack vectors, such as fine-tuning, overwriting, or adversarial attacks, these are avenues for future work. We believe this foundational step represents significant progress, particularly as most existing methods primarily focus on robustness against other types of attacks, such as fine-tuning.

* **Focus on Robustness Against Adversarial Attacks**: Several reviewers raised concerns about robustness against adversarial attacks. However, since our proposed method creates fingerprints using random input-output pairs, adversarial attacks are challenging to execute. Specifically, due to the vast input-output space of LLMs, it is extremely difficult for the attacker to estimate candidate fingerprints. Thus, attacks such as overwriting attacks and membership inference attacks are not practical in real-world scenarios.

* **Novelty and Contribution**: The reviewers' comments primarily highlight concerns about the simplicity of our approach and the limited scope of comparisons. However, as there are currently few methods tailored to resilience against model merging, we believe our proposal offers a novel and timely contribution. Simplicity and effectiveness are the cornerstone contributions of our work.

If you find our responses compelling, we kindly request that you reconsider your evaluations. Additionally, we welcome any further questions or clarifications that may help address your concerns. We would also be grateful for any additional comments or suggestions you might have to enhance the quality of our work.

Thank you once again for your thoughtful feedback and engagement.

Sincerely,

The Authors

---

### Meta-Review · Area_Chair_fdZa · 2024-12-19

**Metareview:**

This paper proposed a fingerprint method for model merging. It is an important topic to the domain. However,  all reviewers still have concerns for this paper. After rebuttal, reviewers still concern about the robustness of the method. AC read all reviewers and rebuttal and agreed with reviewers' concern. AC hopes the authors can improve the paper based on reviews.

**Additional Comments On Reviewer Discussion:**

Reviewers pointed out the robustness of the methods and authors did not provide additional response.

---

### Decision · Program_Chairs · 2025-01-22

Reject